# Biophysics of Voice Onset: A Comprehensive Overview

**DOI:** 10.3390/bioengineering12020155

**Published:** 2025-02-06

**Authors:** Philippe H. DeJonckere, Jean Lebacq

**Affiliations:** 1Federal Agency for Occupational Risks, 1210 Brussels, Belgium; 2Neurosciences Institute, University of Louvain, 1200 Brussels, Belgium; jean.lebacq@uclouvain.be

**Keywords:** vocal onset, glottal attack, soft onset, hard onset, coup de glotte, intraglottal pressure, turbulence

## Abstract

Voice onset is the sequence of events between the first detectable movement of the vocal folds (VFs) and the stable vibration of the vocal folds. It is considered a critical phase of phonation, and the different modalities of voice onset and their distinctive characteristics are analysed. Oscillation of the VFs can start from either a closed glottis with no airflow or an open glottis with airflow. The objective of this article is to provide a comprehensive survey of this transient phenomenon, from a biomechanical point of view, in normal modal (i.e., nonpathological) conditions of vocal emission. This synthetic overview mainly relies upon a number of recent experimental studies, all based on in vivo physiological measurements, and using a common, original and consistent methodology which combines high-speed imaging, sound analysis, electro-, photo-, flow- and ultrasound glottography. In this way, the two basic parameters—the instantaneous glottal area and the airflow—can be measured, and the instantaneous intraglottal pressure can be automatically calculated from the combined records, which gives a detailed insight, both qualitative and quantitative, into the onset phenomenon. The similarity of the methodology enables a link to be made with the biomechanics of sustained phonation. Essential is the temporal relationship between the glottal area and intraglottal pressure. The three key findings are (1) From the initial onset cycles onwards, the intraglottal pressure signal leads that of the opening signal, as in sustained voicing, which is the basic condition for an energy transfer from the lung pressure to the VF tissue. (2) This phase lead is primarily due to the skewing of the airflow curve to the right with respect to the glottal area curve, a consequence of the compressibility of air and the inertance of the vocal tract. (3) In case of a soft, physiological onset, the glottis shows a spindle-shaped configuration just before the oscillation begins. Using the same parameters (airflow, glottal area, intraglottal pressure), the mechanism of triggering the oscillation can be explained by the intraglottal aerodynamic condition. From the first cycles on, the VFs oscillate on either side of a paramedian axis. The amplitude of these free oscillations increases progressively before the first contact on the midline. Whether the first movement is lateral or medial cannot be defined. Moreover, this comprehensive synthesis of onset biomechanics and the links it creates sheds new light on comparable phenomena at the level of sound attack in wind instruments, as well as phenomena such as the production of intervals in the sung voice.

## 1. Introduction

Vocal onset can be defined as the process that occurs between the first detectable glottal (vibratory) movement and the steady-state vibration of the vocal folds. The onset of vocal fold vibration is a dynamic phenomenon with a progressive adjustment of the forces acting until a steady state is reached. Three broad categories of vocal onset (or ‘attack’) are generally recognised: soft (or ‘coordinated’), hard and breathy (or ‘aspirated’) [1,2,3]. To some extent, the soft voice onset is the inverted homologue of the voice offset, during which a damped oscillation of the VFs can be observed, unless the voicing is interrupted by glottal closure [3,4]. The traditional videostroboscopic observation, as used clinically, is not suited for analysing the characteristics of the onset phase.

The aim of this article is to describe and analyse the glottal mechanics—studied by different methods—of the normal subject uttering, under standard conditions, a sustained vowel or a syllable that begins with a vowel. This is different from the Voice Onset Time or VOT used in phonetics. The VOT is defined as the time between the beginning or onset of a vowel sound and the release of a stop/plosive (in standard English/p,b,t,d,k,g/) [5]. By combining techniques of imaging with physical methods, it becomes possible to obtain a detailed insight, qualitative and quantitative, into the essential aspects of the vocal onset:Characteristics and systematisation of the different types of onset.Relationship between intraglottal pressure and glottal area.Phase lag of glottal area in relation to intraglottal pressure.Initiation of oscillation.Intraglottal pressure at start of oscillation.Evolution over time of oscillation amplitude.Number of cycles to reach a steady state.Evolution of frequency during the first cycles.

This article provides an overview of the different modalities of voice onset and their specific characteristics, with special emphasis on the biomechanical aspects and particularly the triggering of the oscillation. A link is also made with pitch matching in singers and the role of VFs in the sound onset of brass instruments. This vision is essentially based on in vivo experimentation, and on direct or indirect measurement of the basic involved parameters: the intraglottal pressure, the glottal area, the transglottal airflow, the VF contact and the acoustic sound. This non-invasive investigation of VF dynamics requires a complex instrumentation involving different techniques, frequently applied simultaneously, without interfering with voice production [6,7]:Pressure sensors with adequate frequency response and sensitivity (Millar catheter), a flowmeter (flowglottograph, FGG), with adequate frequency response and sensitivity (Rothenberg’s mask).Glottal area imaging and morphometry techniques (at adequate speed): videolaryngostroboscopy, photoglottography (PGG), high-speed film, videokymography (VKG) (‘single line scan’), Doppler ultrasound.Transglottal electrical impedance for monitoring VF contact: electroglottography (EGG).Audio-recording (sound oscillogram).

The subject for the experiments involving combined parameters was one of the authors, a Caucasian healthy male trained vocalist with extensive experience in voice production experiments, able to control his vocal emissions very well as to pitch and loudness, and for which the parameters relating to the VFs and the vocal tract, in particular, their dimensions, were perfectly known and verified by various imaging techniques.

## 2. Methodology

### 2.1. Vocal Fold Imaging

The video camera used for the present experiments was a Kay HSV 9700 (High-Speed Video, Kay Elemetrics Corporation; Lincoln Park, NJ, USA). A 300-watt xenon lamp is the light source to illuminate the larynx. The digital video signal is transmitted at 384 Mb/s and a 2 s time window is recorded at 2000 frames per second. Single-line scanning (videokymography; VKG) of VF oscillations is an imaging technique based on a special digital camera mounted on a rigid endoscope (90°). In the fast mode, the video camera gives images of a single line selected from the entire image at a rate of 7812.5 line frames/s (European CCIR system: 625 lines/frame). The resolution is 768 × 1 pixel [8]. Successive lines are displayed so that the image shows the vibratory pattern of the small selected part of the VF length through the cycle [9]. High-speed video systems can extract and display the videokymograms (single-line scans) of several selected lines [3,10].

### 2.2. Glottal Area (Light Flux)

The area of the glottis can be derived from a photometric measure obtained by transilluminating the trachea. A photovoltaic transducer in the pharynx detects the light flux (photoglottography, PGG) [6]. The current produced by the photodiode is pre-amplified by a current-to-voltage converter with a linear response up to 2 kHz, and calibration is possible, based on imaging techniques, as explained in previous articles [3,6,11,12].

High-speed film and videokymography give a global vision of the phenomenon, but photoglottography provides the most accurate measure of the glottal area. Photoglottographic signals are actually more accurate than those obtained by processing images from high-speed video or videokymography [6,7,11].

### 2.3. Transglottal Airflow

The time course of the airflow during each cycle (flowglottogram, FGG) [12] can be recorded using a Rothenberg mask and the MSIF2 system for inverse filtering from Glottal Enterprises (Syracuse, NY, USA). The Rothenberg mask is actually a high-speed pneumotachographm, and its properties have been investigated in depth [13,14,15,16,17]. Time delay correction and calibration procedures have been described previously [6].

### 2.4. Translaryngeal Electrical Impedance

The transverse electrical impedance of the glottis is measured by electroglottography (EGG) [6,7] in which a high-frequency current (>100 kHz) is applied to cutaneous electrodes at the level of the glottis. The measured changes reflect the changes in the surfaces of contact of the VF. This does not interfere with the glottal oscillation, thus allowing any phonetic protocol with acoustic control to be studied. Detecting small variations in impedance, as required in this context, critically depends on the detecting electronic circuitry. As shown in previous work [4], the sensitivity of the EGG signal can be as high as that of the flow signal for detecting very small oscillations of the VFs, but, contrary to the flow signal, it may not detect the very first movements because there is no contact between the VFs. Improved devices show small sinusoidal EGG cycles before true contact occurs over the full VF length [18,19,20]. These small (reduced amplitude) sinusoidal EGG cycles probably correspond to small periodic impedance fluctuations at the level of the ventral commissure.

### 2.5. Doppler Ultrasound [11,21,22]

The oscillating VF reflects the ultrasound signal with a ‘Doppler shift’ dependent on the velocity of vibration. Its main advantage is that it is completely non-invasive and very sensitive. A clean signal is readily detected during the glottal cycle of oscillation, but the shape of the obtained waveform critically depends on the exact position of the Doppler probe facing the glottis, and it is nearly impossible in practice to reproduce it in successive recording sessions. Moreover, the anatomical surfaces reflecting the ultrasound beam cannot be determined, and thus the signal cannot be calibrated in terms of glottal movements.

### 2.6. Acoustic Signal

For combined synchronised recordings, a tiny condenser microphone (Ø 5.6 mm) can be inserted inside the Rothenberg mask laterally: it fits exactly into a small opening of the mask on the opposite side of the pressure transducer. A small delay correction must be applied, in the same way as with the flow signal. The speech samples are processed for sound analysis with the Praat software version 6.4.01 (www.praat.org, accessed on 1 December 2023) [21]. A Wärtsilä 7178 sound level meter served for calibration in a position that corresponds to a direct measurement at 10 cm from the lips.

### 2.7. Intraglottal Pressure

Direct experimental measurement of this essential parameter cannot meet the requirements of non-invasiveness and non-interference with spontaneous voice production.

However, if a synchronised display of glottal area and airflow curves is available, the cyclic velocity of the air particles can be easily calculated. The intraglottal pressure P during the open phase of the vibration cycle can be calculated from the transglottal flow and the velocity of the air particles (flow/area) according to the law of conservation of energy (Bernoulli):P + ½ ρ v^2^ = constant (1)
in which ρ is the density of the fluid and v is the velocity of air particles [12,22,23].

This approach was inspired by a graphic model proposed by Titze in 1988, who showed how the air particle velocity at the glottal level can be obtained by dividing the airflow waveform by the glottal area waveform. In turn, the waveform for the intraglottal pressure can then be inferred using Bernoulli’s energy law [22,23]. In this way, the intraglottal pressure can be easily correlated with other relevant phonatory parameters (raw or after differentiation).

It is easy to demonstrate that, when there is a skewing of the airflow curve to the right with respect to the area curve, the intraglottal pressure is higher during the opening phase than during the closing phase. This skewing is a result of the air compressibility and the vocal tract inertance [3,12,24].

The technical details of these instruments and techniques are described in detail in two recent articles [6,7], as are the methodological issues that are critical to the valid calibration and synchronisation of these signals.

## 3. Systematisation of the Voice Onset Categories

As demonstrated by PGG, EGG, FGG, videolaryngoscopy, high-speed filming and VKG, VF vibration can start either from a glottis that is closed or from a glottis that is open (Figure 1). The basic difference is the presence of airflow before the oscillation starts. In running speech, when the oscillations start from a closed glottis, e.g., when the first word of the sentence begins with a vowel, the adduction force of the VFs is low, comparable to that of the subsequent sustained oscillation. However, most frequently a slightly spindle-shaped glottal slit is observed just before oscillation starts. In this case, airflow can be detected but is normally not audible. Such onsets may be called ‘soft’, and they frequently occur in modal running speech (coarticulation). Hence, as a physiological, soft onset can occur from either a closed or an open glottis, it may be proposed that soft onsets be named soft _(o)_ (from open glottal slit) and soft _(c)_ (from closed glottis), respectively. A voice onset can also be ‘breathy’ when the sound production is preceded by a voluntary and conscious airflow increase, which can in some cases become audible: sometimes the misleading term ‘aspirate’ is used. On the other hand, a voice onset can be ‘hard’, which means that the VF oscillation is preceded not by just a VF-contact on the midline with a low adduction force, but by a stronger closure of the larynx, resembling to some extent an effort closure like in a cough. This is frequently observed in loud and convincing (or exuberant, or angry) speaking, e.g., in actors’ voices. In more extreme cases, the French term ‘coup de glotte’ is used to qualify this effortful, hyperkinetic vocal behaviour. The other extreme is a clearly audible breathiness during the onset, but also to some extent during the sustained sound, which brings the voice closer to whispering. In a soft/breathy voice onset, the number of cycles that can be observed depends on the physiological characteristics of the onset, but also on the physical properties of the recording technique.

In his seminal study, Fink [25] first explained the basic concept of the spindle-shaped glottal configuration and the paramedian axes of vibration of the VFs:

When the onset of phonation is filmed at 16 frames/s, the medial glide of the arytenoids from the respiratory position to their contact on the midline takes approximately 250 ms. In the initial position (usually referred to as the ‘respiratory position’), the median edges of the VFs are straight and join the vocal processes of the arytenoids at an angle of about 15°. During and especially at the end of adduction, the VFs are bowed outwards, before the onset of vibration blurs the image (Figure 2). This outward curvature produces a distinct spindle shape. The essential observation, already made by Fink [25], is that the vibration starts from a paramedian arc and not from the midline, and that the axis of oscillation remains paramedian in the first cycles. With an axis of oscillation on the midline, the VFs would move at maximum speed at the moment of midline contact and the closed quotient (duration of the closed phase/total cycle duration) would be 0.5, which is never observed in a physiological onset phase. The essential concept of the paramedian axis of oscillation, i.e., that the adducting/adducted VFs tend to be bowed outwards (concavity), has been emphasised by Fink and Demarest [26].

Figure 2 shows two consecutive images from Fink’s 16 frames/s film: the left picture is still sharp, and points out the immobile spindle-shaped configuration, while in the right picture (taken 62.5 ms later) vibration has started, evidenced by blurring. Fink also notices that, at higher speed (1000 frames/s), the starting axis constitutes the approximate axis of the VF vibration. Fink logically concludes that paramedian oscillation axes of the VFs actually reduce the trauma due to midline collision. The question of collision speed has been exhaustively explored in our previous work [11].

Figure 3 and Figure 4 are extracted from a videolaryngoscopic recording of a normal male trained vocalist uttering physiological sustained /a:/ ‘s. In each figure, the time interval between the left and the right images is 30 ms. The typical (sharp) pre-onset spindle-shaped configuration appears clearly in both left images (soft _(o)_ onset). In both right images, blurring indicates that VFs are vibrating. The size of the ellipsoidal split may differ to a large extent, as can be seen in Figure 5: two images preceding the oscillation by 30 ms. This difference depends on many parameters (e.g., the targeted loudness), still insufficiently investigated.

On the contrary, Figure 6 is an example of a physiological soft onset, but starting from a closed glottis (soft _(c)_): the VFs are completely (but gently) adducted, without ‘effort closure’ that should tend towards a sphincteric behaviour, which—when present—can be seen from the adduction of the ventricular folds. In the next frame, the VF edges appear blurred, similar to the soft _(o)_ examples.

Figure 7 shows two examples of pre-onset configurations that represent a (moderate) exaggeration of the soft _(o)_ and soft _(c)_ patterns, respectively: on the left is the starting position of a breathy onset, with a significantly larger glottal area, and on the right is the starting position of a clearly hard onset, with concomitant adduction of the ventricular folds.

In summary, the onset pattern shows a wide range of modalities, achieving a continuum from the almost whispered onset to the ‘coup de glotte’, with between these extremes the breathy, soft _(o)_, soft _(c)_ and hard onsets. However, the only quantitative bicategorial distinction is based on the open or closed glottis at the moment of vibration onset, and subsequently the presence or absence of transglottal airflow. Clinicians usually consider only the extremes to be abnormal (at least if recurrent/habitual), except when deliberately used for expressive purposes, e.g., by actors. A ‘coup de glotte’ may be considered phonotraumatic, and a whispered onset may be considered energetically demanding and therefore a source of vocal fatigue.

The anatomical substratum for the essential concept of the paramedian axis of vibration, i.e., the tendency of the adducting/adducted VFs to be curved outwards, consists of interstitial elastic fibres whose influence is evident in the curved medial borders of the adducted folds prior to onset at modal frequencies [25,26,27]. Moreover, this pre-phonatory adaptation also has a muscular component, as indicated by the increased electrical activity in the vocalis muscle just before the onset of vibration [28,29,30].

## 4. Morphological Analysis

### 4.1. Imaging

In a normal subject, the most commonly observed type of voice onset in spontaneous speech is the soft _(o)_ onset, which deserves more in-depth analysis. A typical soft (slightly breathy) onset in modal voicing conditions, as observed on four videokymograms (four different glottal levels) obtained from a high-speed video (2000 frames/s), is shown in Figure 8: the VF oscillation is initiated from a spindle-shaped glottis (Figure 9). It is not possible to determine whether the initial movement is medial or lateral. As long as there is no contact between the VF edges, the amplitude of the oscillations very progressively increases. The first contact between the VFs on the midline is very short but subsequently, the duration of the contact phase increases progressively. This is also visible in the very first cycles on a VKG at the approximate midpoint of the glottal length (Figure 1, left image).

### 4.2. Polygraphic Recordings

How a soft and a hard onset differ is clearly visible in Figure 10 and Figure 11. The essential difference is whether the glottis is closed or open just before the first oscillation, and whether or not there is a transglottal airflow.

#### 4.2.1. Onset with Closed Glottis (Soft _(c)_/Hard)

In Figure 10, the glottis is closed (from top to bottom: flowglottogram, EGG, photoglottogram, sound) before the onset of oscillation: no airflow, no sound, EGG indicating contact between the VFs, and no transglottal light. Sound occurs with the first glottal opening and the duration of the closed phase increases progressively. With a hard or soft _(c)_ onset (Figure 10), the amplitude of the oscillations also progressively increases, but generally, the number of cycles to reach a steady state is less than with a soft _(o)_ or breathy onset.

When phonation begins with a closed glottis (VFs fully adducted, no airflow), the subglottic pressure increases until it is sufficient to overcome the glottal resistance; mechanically, it is the sudden imbalance between the intratracheal pressure and the muscular tension of the adductor muscles that initiates the oscillation [28].

#### 4.2.2. Open Glottal Onset (Soft _(o)_)

In a soft _(o)_ onset (Figure 11) in comfortable conditions (125 Hz; 65 dB at 10 cm), the FGG (flow trace) first detects an oscillation, immediately followed by the area trace. EGG (vocal fold contact) changes appear later (Figure 11). A closed plateau only clearly appears after the fourth cycle. The EGG trace shows that a very short and limited contact occurs in the previous cycle.

In a breathy onset (Figure 12), the pattern is similar to that observed in a soft onset, but it develops more slowly, the oscillation amplitude progressively rising during at least ten cycles.

## 5. Biomechanics

### 5.1. Relationship Between the Intraglottal Pressure and the Glottal Area [3] 

Applying the Bernoulli equation (Equation (1), the pressure at the level of the VFs (P) can be computed by combining the transglottal air flow and the air velocity (i.e., air flow divided by glottal area) on the basis of the aforementioned Bernoulli equation (Equation (1)) [12,22,23] according to a model proposed graphically by Titze, who also discusses in his article of 1988 [22] the relation with the original model of Ishizaka and Matsudaira, who analysed the pressure–flow relationships in a glottal duct modelled using two masses and three springs. These considerations apply to sustained phonation. The glottal duct takes indeed—during the open phase—in sequence, a convergent, uniform and divergent shape, as shown and discussed in previous work [24].

However, when the glottis is open, the overall pressure distribution in the glottis is modified as the supraglottal acoustic pressure affects the intraglottal pressure [12,24].

In fact, Equation (1) is applicable when the glottal duct is convergent along the vertical dimension, meaning that it is narrower on the downstream side. However, for a divergent glottal duct, meaning the duct is narrower on the upstream side, where airflow separation from the wall and vortices could occur, it is necessary to consider the inertance equationP = IdU/dt (2)
where I is the supraglottal acoustic inertance and U is the airflow.

The inertia of the mass of the air column above the glottis creates a reacting force to opposing air accelerations, thus the progression of the wave of oscillation beyond the glottis. This inertia, or reactance, can be calculated by Equation (3):I = ρL/S (3)
in which S is the cross-sectional area of the air column and L is its effective length [31].

Its SI units are kg.m^−4^. It can be assumed that L and S remain constant throughout the production of a sustained vowel, as is the case in these experiments. However, such a divergent shape of the glottal duct is primarily observed at elevated subglottal pressures, wherein the VF vibration cycle is distinguished by a prolonged closed phase and a notable phase difference between the lower and upper margins of the VFs. During onset, the subglottal pressure is low and the vertical glottal duct is expected to be shorter [24,32]. Also, the (convergent/divergent) shape differentiation—including the effects of separation of airflow from the glottal wall and formation of vortices—is expected to be less pronounced than is the case in sustained modal voicing, or even absent. During a soft _(o)_ onset, there is no closed phase, which means that Equation (1) permanently intervenes and that Bernoulli’s energy law predominates [24]. It may therefore be reasonably assumed that—in contrast to sustained phonation—the average driving pressure (from bottom to top) is approximately equal to the Bernoulli pressure, which can be numerically estimated via the glottal area at the place where the glottis is the narrowest [24]. It can be demonstrated that when there is a skewing to the right of the airflow curve with respect to the glottal area curve, the intraglottal pressure is higher during the opening phase than during the closing phase. This skewing is a result of the compressibility of air and vocal tract inertance [24]. In the case of a true incompressible fluid, vocal folds would not be able to vibrate.

Thus, the time course of P, the intraglottal pressure, can be approximated by applying Equation (1). An example of the resulting waveform is shown in Figure 13. The expected phase lead of roughly 90° between the pressure and the glottal opening is effectively observed. This can be attributed mainly to the airflow curve being skewed to the right relative to the glottal area curve. In Equation (2), P is a function of the first derivative of the flow, resulting in a phase lead of 90° relative to the flow. This results in P during the opening phase of the cycle being larger than during the closing phase, from the initial onset cycles onwards, which is required for a transfer of energy from the lung pressure to the mass of the VF.

### 5.2. Time Course of the Relationship Between Glottal Area and Intraglottal Pressure

The evolution of the phase lead of the intraglottal pressure relative to the glottal opening is shown in Figure 14. The figure illustrates the increasing phase lead of the pressure curve—predominantly determined by Equation (1)—relative to the glottal area curve in successive cycles during the soft _(o)_/breathy onsets [24]. The data are represented as means, standard errors of the mean and standard deviations.

Twenty-eight onsets are initially considered, but as the sequential number of the cycle increases, the number of cycles decreases. As the onset progresses, the intervals between the glottal area peak and the intraglottal pressure peak increase. The number of cycles in the 28 individual recordings suitable for this computation ranged from three to nine. The mean phase lead progressively rises from 0 to about 0.9 ms, equivalent to 10% of a cycle. In an ideal undamped frictionless harmonic oscillator driven by an external periodic force, the force should lead the displacement by 90°. This is a theoretical case. The discrepancy between our observations and the ideal case is likely due to frictional forces in the real system. In sustained phonation, the pressure wave lead varies from 55 to 85°. This finding is in line with published qualitative observations [33]. Moreover, the observed skewing of the area curve to the right with respect to the flow curve increases with loudness in sustained phonation. This trend is likely to be replicated in onset [12,24].

### 5.3. Initiation of the Vibration in a Soft _(o)_/Breathy Onset

In vivo records of glottal area and airflow show that the oscillation starts precisely at the emergence of turbulence in the narrowed glottal air flow, corresponding to the critical value of the Reynolds number [34].

Figure 15 shows an example of a record during a soft _(o)_ onset (from top to bottom: airflow, EGG, the glottal impedance and PGG, the glottal area). The record duration is 124 ms. On the area trace, the level at which oscillation begins and the maximal amplitude (100%), are marked by vertical arrows. Calibration is made on images. On the airflow curve, oscillation begins at a time marked by a vertical arrow. The baseline (flow = 0) is the value of flow at complete glottal closure. The calculated values of Reynolds numbers in 72 experiments range between 2700 and 3100 [34].

Turbulence in the airflow can be assumed to be the triggering factor in an oscillator including the mass of the VFs and of the air column in the vocal tract. It is admitted that this system is weakly damped [4,7]. Turbulence can act as a specific trigger of the oscillator, particularly at a time when the subglottal pressure is low and the vertical glottal duct is short before any convergence/divergence appears [6,24,32]. The frequency of oscillations is then determined by the mechanical parameters of the system.

Figure 16 depicts a plot of the equivalent glottal diameter (in mm.) against the velocity of air particles (in metres per second) at the onset of oscillation. The equivalent glottal diameter has been calculated from the measured glottal area. A strong negative correlation (R = −0.80; *p* < 0.0001) is observed, and the hyperbolic shape of the regression curve suggests that the velocity varies inversely with the equivalent glottal diameter. The approximately constant product corresponds to a Reynolds number of around 3000. It should be noted that the size of the ellipsoidal split may differ considerably, as illustrated in Figure 5, which depicts two images taken 30 ms before the onset of oscillation.

### 5.4. Intraglottal Pressure at the Time of Start

With all methods used (photoglottography, flow glottography, ultrasound glottography), the first glottal oscillations always appear to be nearly symmetrical around their oscillation axis (Figure 8). Imaging methods, such as videokymography and high-speed video, which show the two VF separately, also demonstrate that the two axes of oscillation remain equidistant and that they are neither pulled medially nor pushed laterally. The VF movements are symmetrical (adduction/abduction) on either side of a neutral position.

The VFs oscillate freely, without exhibiting gross suction between the two sides (Bernoulli effect) or any significant separation driven by the airflow and lung pressure (blown apart). This indicates that the pressure at the glottal level is approximately equal to the atmospheric level at the onset of oscillation. To substantiate this perspective, it is essential to demonstrate that the opposing forces are in equilibrium at the pivotal moment. The results of our calculations indeed provide comparable mean values for both positive lung pressure (2.52 +/− 1.58 hPa) and negative Bernoulli force (2.19 +/− 1.26 hPa) [7,34]. While some of the measurements (glottal area, estimate of lung pressure) may lack precision, it is reasonable to assume that the overall error does not exceed 20% [34]. At the critical time, the rising positive intratracheal pressure is balanced by the rising negative Bernoulli pressure generated by the transglottal airflow.

### 5.5. Time Course of the Amplitude of Oscillation

There is a considerable variation in the number of cycles of a vocal onset. In general, the amplitude of oscillations, as measured on the photoelectric signal, increases gradually over two to more than thirty cycles before reaching a first clear closed plateau. Plots of the increase in amplitude of glottal area peaks typically exhibit a sigmoid shape (Figure 17). Upon contact, the sinusoidal signal is clipped, and the amplitude exhibits a slight decrease. A comparable pattern is observed for the flow peaks (flowglottogram).

The initial cycles observed in the EGG signal are sinusoidal, whereas subsequent cycles exhibit a peakier shape (see Figure 11 and Figure 12). In a hard onset, the electrical impedance undergoes a change first (downward movement), indicating a reduction in vocal fold contact area despite the glottis remaining closed. This is accompanied by an increase in impedance, which suggests a glottal opening. A closed plateau is present from the first cycle onwards.

Figure 11 also shows that the ultrasound signal is as sensitive as the other signals in revealing initial VF movements. The ultrasound wave is clearly related to glottal movements, but the relation between its peaks and the opening or closing of the glottis varies largely between recordings. Thus, the direction of the ultrasonic beam relative to the moving parts of the larynx cannot be controlled, so structures reflecting the ultrasonic beam during oscillation cannot be identified. This considerably limits the possibility of using this method.

### 5.6. Number of Cycles Before Steady State

In modal phonation conditions, the number of discernible cycles of oscillation preceding the steady state is highly variable. A larger number of cycles is observed in the onset phase of the ‘soft _(o)_ /breathy’ condition than in the ‘soft _(c)_ /hard’ condition (open vs. closed glottis). With respect to the sensitivity of the different techniques (photo-, flow- and electroglottographic traces), the results of the ANOVA indicate a significant difference between them (*p* < 0.001, n = 59) (Figure 18). This is evidenced by the observation that fewer cycles are observed on the EGG traces (approximately four), while a similar number of cycles (approximately seven) are detected in the flow and area traces. The considerable variability appears to be attributable to a number of factors [3]. Two factors can be identified from the tracings themselves: the VF adduction velocity and the peak flow at the end of the onset phase. The VF adduction velocity is defined as the slope of the glottal area signal immediately preceding the onset of oscillation. When the aforementioned slope is sufficiently stable to be reasonably approximated, a weak but significant negative correlation emerges between the number of cycles and the adduction velocity [3]. It stands to reason that when adduction is faster, the onset process is shortened. Conversely, when the expiratory flow is higher, for instance, in an intentionally breathier onset, the number of pre-steady-state cycles significantly increases [3].

### 5.7. Evolution of Frequency During the First Cycles

In instances where a soft _(o)_/breathy onset is observed, and a sufficiently large number of cycles are present, the fundamental frequency of the VF oscillations tends to slightly decrease progressively, concomitantly with an increase in the vibrating mass. Figure 19 illustrates the progress of cycle duration over the initial 18 cycles of a breathy onset, prior to the attainment of a closed plateau [3]. This suggests that when the vibrating mass is confined to a narrow strip of tissue along the VF edge, the vibration frequency is higher than when a larger mass of the VF is involved.

## 6. Frequency Control in Onset of Singing Intervals

It is established that voice onset exhibits a significantly greater degree of fundamental frequency perturbation (jitter) than the steady-state midportion of a sustained vowel [35]. Figure 20 illustrates the period duration of the initial 10 cycles of a sustained /a:/ in a typical male subject (10 repetitions of a soft _(o)_ voice onset). In instances of hard onset, period irregularities are often observed in the initial cycles.

In an original test of pitch-matching with physiological onsets [30], ten young female conservatory singing students (ages 18–22) were compared with their female singing teachers (ages 26–43) in a vocal pitch-matching task consisting of producing three standardised ascending intervals (a third, a fifth and an octave, starting from the same fundamental base note d1, 294 Hz) and in two conditions (with/without modelling by a piano). The subjects were instructed to sing on the vowel /a/ with a pleasant volume without vibrato and with a short interval between the two notes of each interval (so no legato and no portamento) (Figure 21). The experiment was carried out as part of an ordinary singing course, and in a class at the Conservatoire. All students and teachers agreed to participate. The period duration at the offset of the base tone and at the beginning of the target tone was analysed in detail. The statistical comparison of the F_0_ quotients for the three intervals shows that, unlike untrained subjects, singing students have already reached a level of neuromuscular control in pitch adjustment comparable to that of real professionals. In addition, the extent of the interval does not affect the accuracy of pitch adjustment, and the reference to the piano is not a relevant factor. In general, F_0_ instability is observed in the first cycles of the target tone, which could be due to mechanical muscle tension readjustments. Both the student and teacher groups show a significantly higher degree of F_0_ perturbation (coefficient of variation in the period) in the first five cycles of the second (target) tone compared to the following ten cycles (*p* < 0.001). This increase is observed in all intervals and under all modelling conditions. The size of the interval has no influence on this result. Moreover, it is remarkable that the periodicity of the last cycles of the base tone is significantly disturbed precisely at the moment when the subject anticipates the pitch jump, this perturbation increasing with the degree of pitch jump to be achieved.

The underlying neuromuscular phenomenon is possibly something equivalent to the pre-phonatory burst of asynchronous muscle action potentials observed in the VF muscles before the onset of phonation [28,29]. However, the phenomenon occurs so rapidly that it cannot be perceived by the listener.

The hypothesis that professional classical singers have a more accurate neuromuscular control in pitch tuning (musical intervals) than young singing students seems thus to be rejected. The modelling by a piano of the tone intervals to be produced seems also a not relevant factor.

As anticipated, the frequency perturbation is, for all conditions, significantly larger during the initial cycles of the target tone than in the subsequent ten cycles. This phenomenon may be attributed to a necessary mechanical readjustment and stabilisation of muscle tension, as well as a new equilibrium between glottal resistance and subglottal pressure [23]. No discernible difference is observed between groups or between intervals.

A very intriguing aspect is the period perturbation that seems to be induced by the anticipation of the pitch jump. The jitter phenomenon remains incompletely understood. It is typically explained by mechanical factors within the layered VF tissues (viscosity, non-linearity), by internal body noise (blood pulsations) and by asynchronic firing of motor units in the M. vocalis and cricothyroideus. This asynchronic firing, particularly in the M. vocalis, is likely to be primarily responsible for the increase in F_0_ perturbations observed in this experiment. The pre-phonatory tuning of the M. vocalis is a well-documented phenomenon that can be clearly observed using electromyography [28,29]. The thyroarytenoid muscles exhibit a burst of muscle action potentials that precede the onset of voicing, followed by a decrease in activity [36]. The interval between the start of the change in electrical activity and the initiation of the vocal emission is about 100 to 200 ms (actually from 50 to 600 ms). This enhanced pre-phonatory activity is characterised by the firing of motor units within the M. vocalis, and the rate of firing is outside of the VF vibration frequency range. This could explain the perturbation effect on the regularity of this vibration frequency. As it is known that in the modal register there is a clear positive correlation between voice pitch and muscle tension in the M. vocalis [29,37], it may be expected that the importance of such a pre-target tuning activity is dependent on the extent of the ascending tone interval. Consequently, it may be hypothesised that the interference with the muscular tension in the last ms of the base tone also depends on the extent of the interval.

## 7. VF Intervention in Sound Attack of Brass Playing

A clear analogy can be drawn between the biophysics of VFs, particularly with regard to the onset of phonation, and the functioning of another biological oscillator: the lips in brass playing. The involuntary and unconscious participation of the vocal folds during brass playing has been empirically verified, particularly during the onset of sound [38]. In the conventional technique of brass playing, the sound is produced at the level of the lips of the performer.

The EGG [1,6], which monitors the VF contact without interfering with the playing of the instrument, has demonstrated that in professional instrumentalists (horn players), the soft attack of the audible and recordable sound, which corresponds to the start of lip vibration, is frequently preceded by one to fifteen glottic oscillations at the frequency of the sound to be generated at lip level. However, this phenomenon can only be observed when the lips are in a position that is neither in contact nor close together, and when the tongue does not obstruct the passage of expiratory air (see Figure 22). Since Damsté (1966) [39], lip mechanics have been extensively investigated, particularly by using transparent mouthpieces, which allow for stroboscopic and high-speed film recordings.

An illustrative example can be observed in Figure 23, which depicts a simultaneous recording of an electroglottographic (EGG) signal and a sound oscillogram (SO) in an attack of a sound at 175 Hz (horn). In the EGG signal, the maximum VF contact is below, while the minimum is above. The emission of sound is preceded by the occurrence of five glottal cycles. The amplitude of glottal movements is observed to be larger during the initial five cycles than during the sustained emission. The glottal frequency of the initial cycles is marginally higher than that of the intended sound, yet the adjustment occurs with remarkable rapidity [38].

It is noteworthy that in a male player, no preliminary glottal vibration occurs at high pitches, such as 580 Hz and 696 Hz (i.e., above the range of the typical male singing voice). Conversely, in a female player, at low pitch (B flat 57 Hz), the preliminary glottal oscillations are present, but at a multiple (three- or fourfold) of the frequency of the note to be played.

The underlying hypothesis is that this strategy allows the brass player to benefit from the fine neuromuscular regulation of the VF muscles, as used in singing, in order to gain precision in adjusting the lip vibration by pre-pulsing the expired airflow at the desired pitch.

Moreover, the spindle-shaped configuration of the VF prior to the onset of oscillation in a soft _(o)_ mode (as illustrated in Figure 2, Figure 3 and Figure 9) also distinctly resembles that of the mouthpiece of double-reed musical instruments, such as the bassoon. Experiments have been conducted with double-reed musical instruments, particularly the crumhorn, as the latter is equipped with a cap that prevents any contact with the player’s lips. The spindle-shaped static configuration (Figure 24) of the double reed enables the precise measurement of an area during a straightforward progressive (monitored by a pneumotachograph) increase in airflow by the player. This allows the exact flow value at which sound is elicited to be determined. The value of the critical Reynolds number consistently corroborates the concomitance of the emergence of turbulence [40].

## 8. Conclusions

Vocal onset can be specified as the process that occurs between the first detectable vibratory movement of the glottis and the steady-state vibration of the VFs. The concept of a ‘critical phase of phonation’ was first introduced in the title of the seminal article by Cornut and Lafon (1959) [41]. Subsequent to their pioneering work based on spectrographic analysis, considerable advancement has been made in the biophysical comprehension of voice onset, due to the integration of imaging and non-invasive physiological techniques, particularly when employed in conjunction. The present article provides an overview of the various modalities of voice onset and a systematic account of their distinctive characteristics, with particular attention to biomechanical aspects. It is crucial to comprehend the temporal relationship between glottal area and intraglottal pressure, which led to our three key findings:

(1) From the initial onset cycles onwards, the intraglottal pressure signal leads to that of the opening signal. This means that the intraglottal pressure during the opening phase of the glottis is systematically greater than that during the closing phase, which is the basic condition for an energy transfer from the lung pressure to the VF tissue, similarly to what occurs in sustained phonation.

(2) This phase lead is primarily due to the skewing of the airflow curve to the right with respect to the glottal area curve. The skewing is a consequence of the compressibility of air and the inertance of the vocal tract

(3) The initiation of the oscillation, when this oscillation starts from an ellipsoidal glottal slit, can be attributed to a shift in airflow dynamics (occurrence of turbulence in a laminar airflow), as evidenced by Reynold’s number calculations. The lung pressure is compensated by the Bernoulli force, and this allows the VF edges to commence their oscillation freely and symmetrically, without either being drawn together by the suction of the airflow or pushed apart by the lung pressure.

The number of cycles preceding the steady state varies considerably (from two to more than thirty), primarily contingent on the type of onset and the aerodynamic conditions. In many cases, the initial cycles exhibit subtle irregularities (beyond the perceptual threshold) even in voices deemed healthy. As the vibrating mass of the vocal folds gradually increases during these initial cycles, there is often a concomitant slight decline in the fundamental frequency (also beyond the perceptual threshold).

It is possible to evoke relevant physical analogies through the use of sound onsets in brass and double-reed instruments. Furthermore, a link is identified between pitch matching in singers and the production of standardised intervals, as well as the occurrence of short-term period irregularities accompanying such pitch jumps.

However, our findings have their limitations: this overview addresses only the biophysical aspects of voice onset in the normal subject, and is focused on modal emissions. The complexity of certain experiments limits the number of subjects. Moreover, our work is purely experimental and of physiological nature, and does not address relations with phonetic, phonological and linguistic phenomena, such as prosody, coarticulation, language acquisition/disorders and cross-linguistic phonetic variations. We just mention possible deviances from the physiological patterns, but our aim is not to consider pathology. We hope of course that researchers in all these fields possibly can use our findings for making fruitful links with their own topics and research questions. In this context, it would also be relevant to elaborate, if possible, a single non-invasive measurement method that would provide reliable, sensitive and specific results applicable to clinical voice patients [42].

## Figures and Tables

**Figure 1 bioengineering-12-00155-f001:**
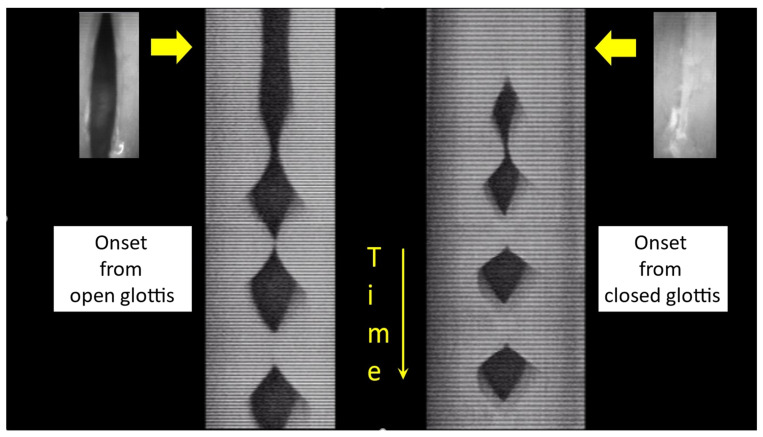
VKG (single-line scan) at approximately the midpoint of VF length, with the related images obtained from high-speed video just before the onset of VF vibration. Total duration about 34 ms. Left: physiological onset starting from an open glottis; right: physiological onset starting from a closed glottis. Hence, there is a wide range of voice onset modalities, but in experimental (physiological) conditions (i.e., on normal subjects, trained vocalists and singers) most authors interested in voice physiology make the distinction between the ‘soft’ (albeit slightly breathy, for clarity) onset, with airflow, and the ‘hard’ onset without airflow.

**Figure 2 bioengineering-12-00155-f002:**
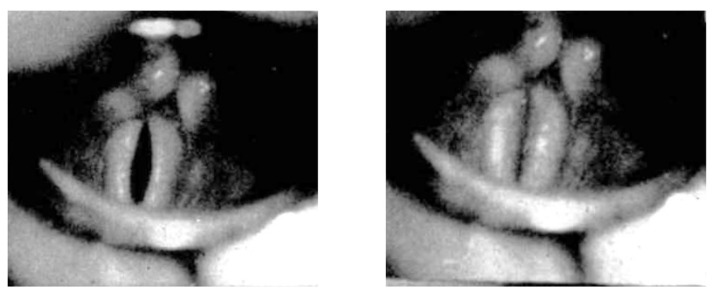
Physiological soft _(o)_ onset: two consecutive images from Fink’s 16 frames/s laryngoscopic film: the picture on the left is still sharp, and points out the immobile spindle-shaped configuration, while in the picture on the right (taken 62.5 ms later) vibration has started, evidenced by blurring. Scale is not indicated on Fink’s film, but the length of the vibrating VFs must be about 15 mm.

**Figure 3 bioengineering-12-00155-f003:**
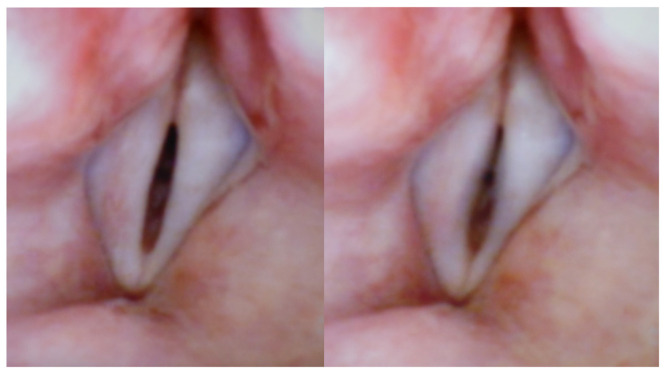
Snapshot extracted from a videolaryngoscopic recording (showing glottis and VFs) in a normal male trained vocalist uttering a physiological sustained /a:/ at 130 Hz (soft _(o)_ onset). The time interval between the left and right images is 30 ms. The typical (sharp) pre-onset spindle-shaped glottal configuration appears clearly in the left image. In the right image, blurring indicates that VFs are vibrating. The length of the vibrating part is 13 mm.

**Figure 4 bioengineering-12-00155-f004:**
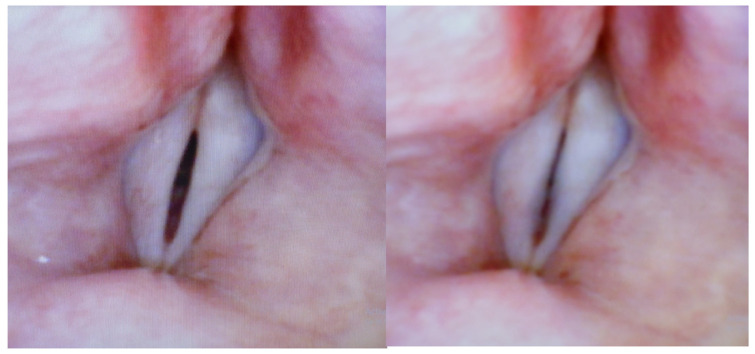
Idem as Figure 3, with a different voicing condition (slightly higher and louder, 145 Hz), and a slightly different pre-onset glottal configuration.

**Figure 5 bioengineering-12-00155-f005:**
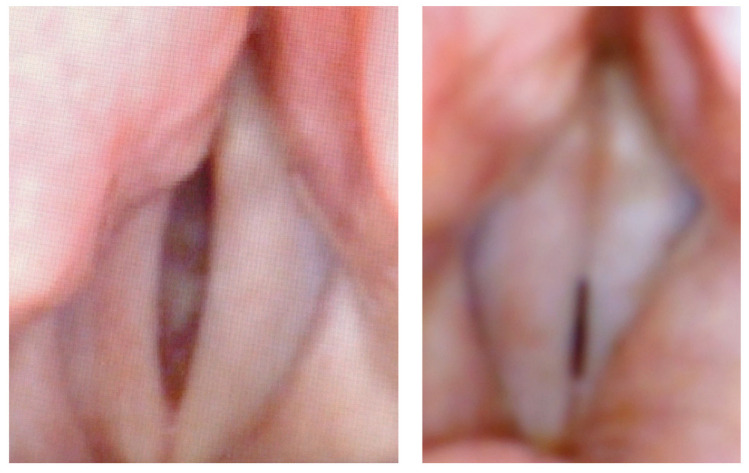
Two images preceding the oscillation by 30 ms, demonstrating how the pre-oscillatory size of the ellipsoid may differ to a large extent.

**Figure 6 bioengineering-12-00155-f006:**
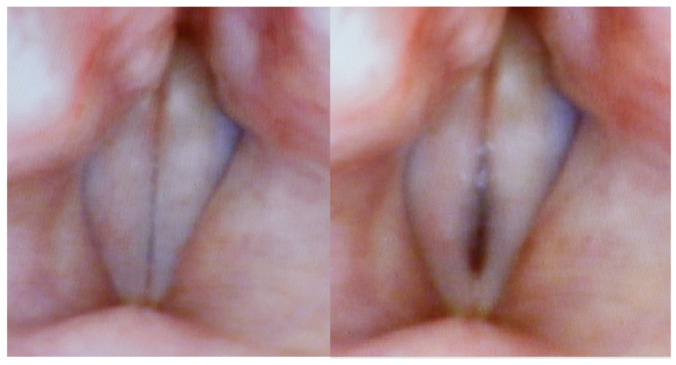
Example of a physiological soft onset (125 Hz), but starting from a closed glottis (soft _(c)_): Left: the VFs are completely (but gently) adducted, without ‘effort closure’ that should tend to-wards a sphincteric behaviour. Right: (next frame, 30 ms later) the VF edges appear blurred, similarly to the soft _(o)_ examples.

**Figure 7 bioengineering-12-00155-f007:**
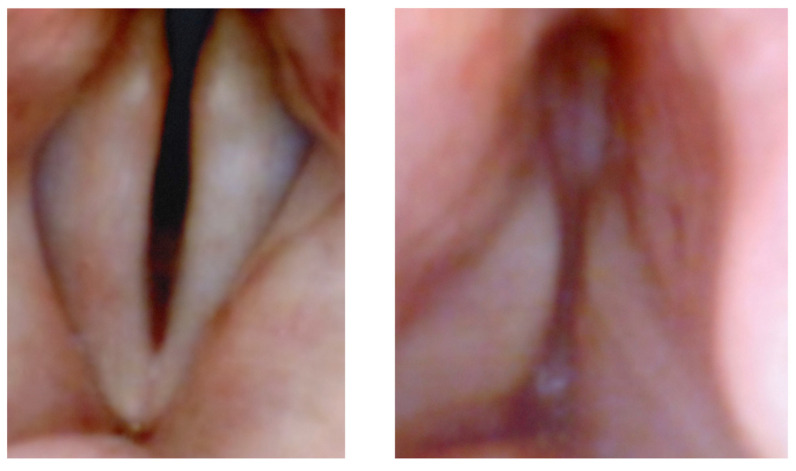
Two examples of pre-onset configurations that represent a (moderate) exaggeration of the soft _(o)_ and soft _(c)_ patterns, respectively: the picture on the left is the start position of a breathy onset, with a significantly larger glottal area, and the picture on the right is the start position of a clearly hard onset, with adduction of the ventricular folds.

**Figure 8 bioengineering-12-00155-f008:**
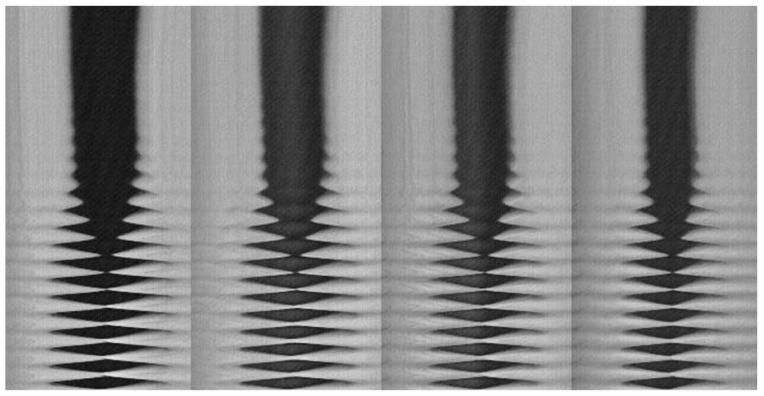
VKG at four equidistant levels of the vibrating glottis, obtained from high-speed video (2000 frames/s). Soft _(o)_, slightly breathy onset. Time is progressing from top to bottom. /a:/; healthy male vocalist (~125 Hz; 65 dB at 10 cm).

**Figure 9 bioengineering-12-00155-f009:**
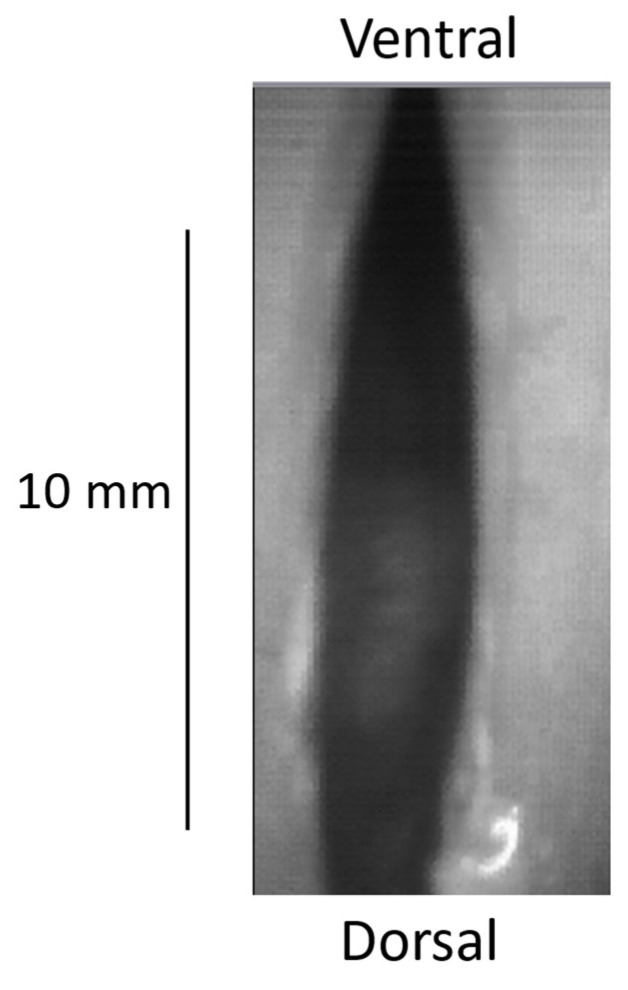
Spindle-shaped glottal split just before the oscillation starts in a soft _(o)_, slightly breathy onset (snapshot from high-speed video 2000 frames/s). Healthy male vocalist (~125 Hz; 65 dB at 10 cm).

**Figure 10 bioengineering-12-00155-f010:**
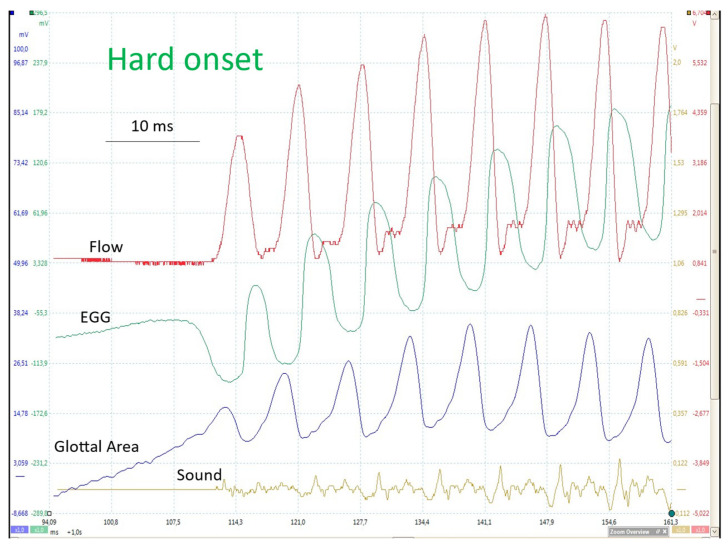
Onset from a closed glottis. From top to bottom: FGG, EGG, PGG and sound oscillogram. Modal phonation (~125 Hz; 65 dB at 10 cm).

**Figure 11 bioengineering-12-00155-f011:**
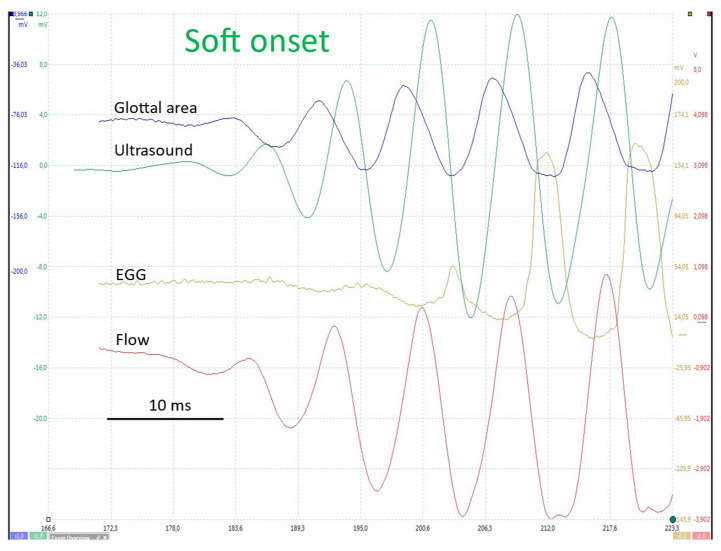
Onset from an open glottis. From top to bottom: PGG (glottal area), ultrasound signal, EGG and FGG as a function of time. Comfortable pitch and loudness (~125 Hz; 65 dB at 10 cm). As can be seen on the glottal area and flow traces, a closed plateau first clearly appears after the fourth cycle. The EGG trace indicates that a very brief and limited contact already occurred in the previous cycle.

**Figure 12 bioengineering-12-00155-f012:**
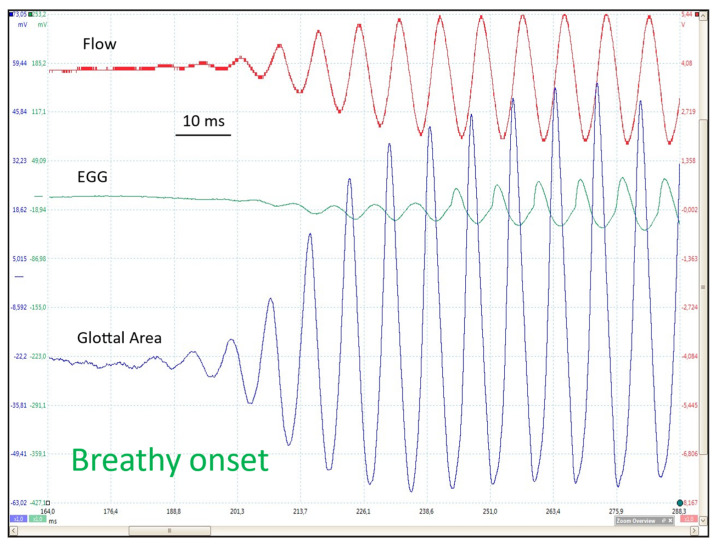
Breathy onset. From top to bottom: FGG, EGG, PGG (raw tracings). Comfortable pitch and loudness (~125 Hz; 65 dB at 10 cm). Similar pattern to a soft _(o)_ onset, but slower progression: the amplitude of oscillations progressively increases over more than 10 cycles.

**Figure 13 bioengineering-12-00155-f013:**
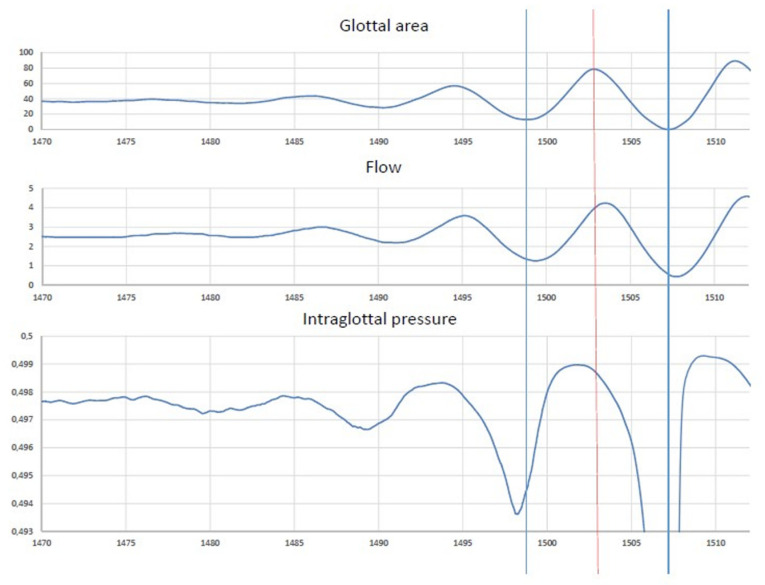
Soft _(o)_ onset (~125 Hz; 65 dB at 10 cm) From top to bottom: PGG, FGG and intraglottal pressure. A phase lead (slightly less than 90°) of the intraglottal pressure with respect to the glottal opening is observed. Time in ms.

**Figure 14 bioengineering-12-00155-f014:**
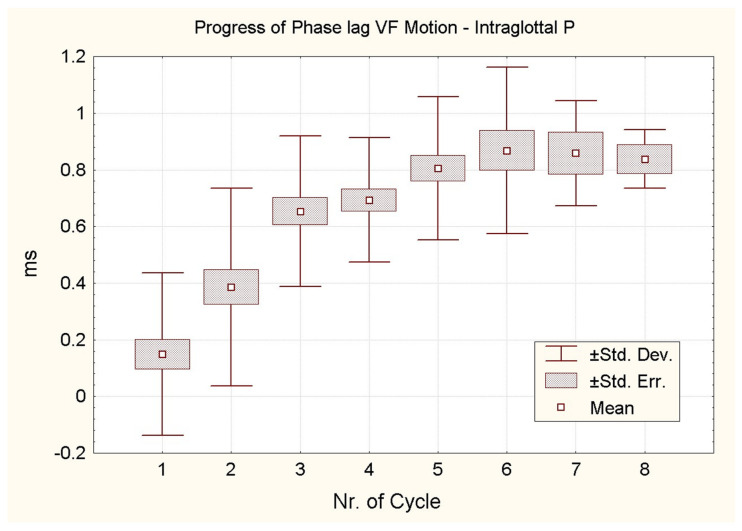
Progress of phase lead (ms) of the intraglottal pressure with respect to the glottal area as a function of the sequential number of the cycle during the soft _(o)_/breathy onsets. The number of cases falls as the number of the cycle increases. The average phase lead of the pressure rises from 0 up to about 0.9 ms.

**Figure 15 bioengineering-12-00155-f015:**
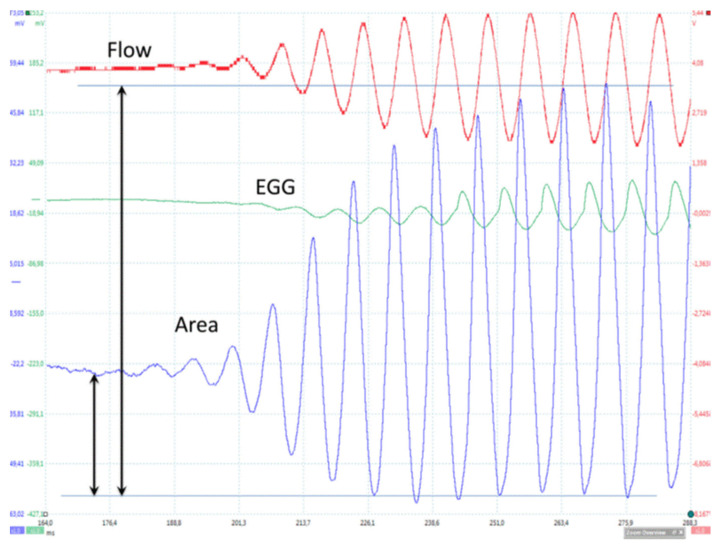
Typical example of a soft _(o)_ onset recording. From top to bottom: FGG (Rothenberg mask), EGG and light signal proportional to the glottal area (PGG). Total duration of the recording is 124 ms. On the area trace, the level at which oscillation starts and the maximal area amplitude (100%)—which can be calibrated on images—are indicated by vertical arrows. The flow level at which oscillation starts is indicated by the vertical arrow on the airflow trace with respect to the baseline (flow = 0) reached when complete glottal closure is observed.

**Figure 16 bioengineering-12-00155-f016:**
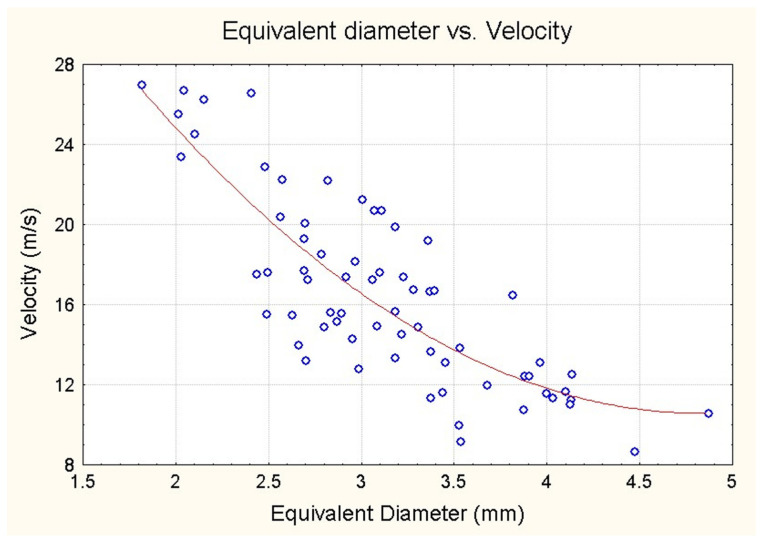
Equivalent diameter (mm) calculated from the measured glottal area as a function of the velocity of air particles (m/s) at the start of oscillation.

**Figure 17 bioengineering-12-00155-f017:**
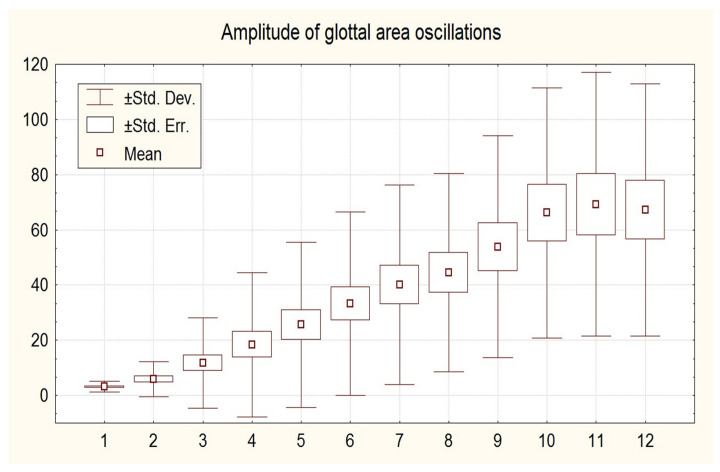
Amplitude (y-axis) of glottal area oscillations during the first cycles (n = 35). Arbitrary units, linear scale. Cycle numbers are on the x-axis.

**Figure 18 bioengineering-12-00155-f018:**
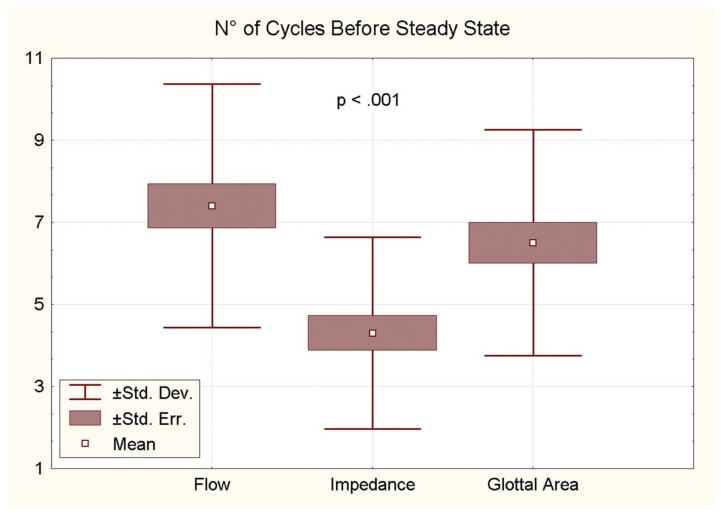
Number of identifiable oscillation cycles before steady-state VF vibration is reached in three different signals: FGG, EGG and PGG.

**Figure 19 bioengineering-12-00155-f019:**
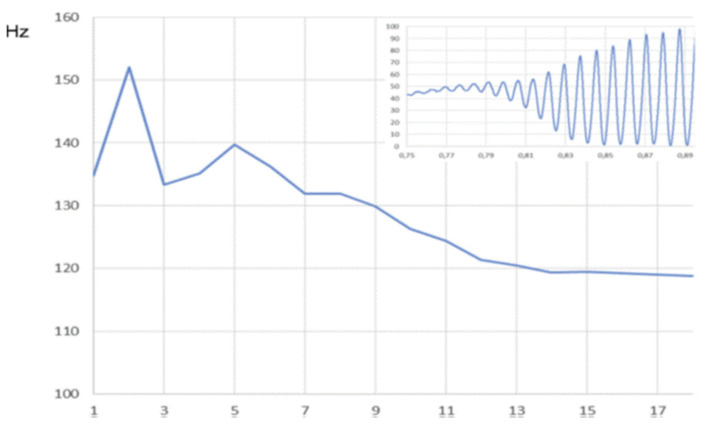
Example of the evolution of cycle duration over the first 18 cycles of a soft _(o)_/breathy onset (125 Hz; 65 dB at 10 cm). Cycle numbers are on the x-axis. The insert shows the glottal area trace. Slight progressive decrease in the fundamental frequency.

**Figure 20 bioengineering-12-00155-f020:**
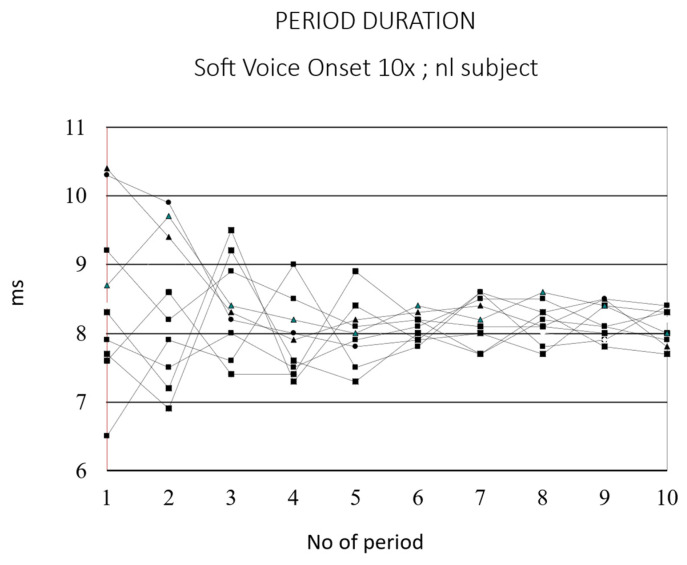
Period duration of the first 10 cycles of a sustained /a:/ in a normal male subject (10 repetitions of a soft _(o)_ voice onset). Period duration becomes stable after about 5 cycles.

**Figure 21 bioengineering-12-00155-f021:**
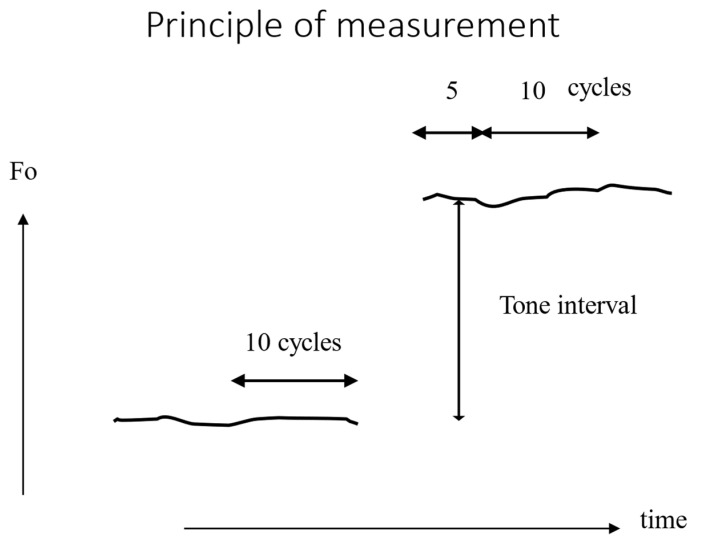
Pitch-matching task for a standardised interval. Principle of measurement: X-axis is time, Y-axis is fundamental frequency. The singer makes a pitch jump (third, fifth or octave) without legato, i.e., with a short interruption in vocal emission. Cycle duration is measured just before (10 cycles) and just after (5 and 10 cycles) the pitch jump.

**Figure 22 bioengineering-12-00155-f022:**
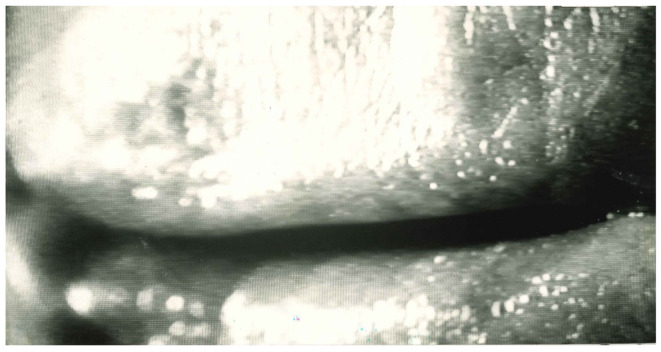
Lip configuration of a professional horn player, as filmed through a transparent mouthpiece. The picture precedes the vibration onset by 50 ms. The lips are slightly parted, and there is a small airflow escape.

**Figure 23 bioengineering-12-00155-f023:**
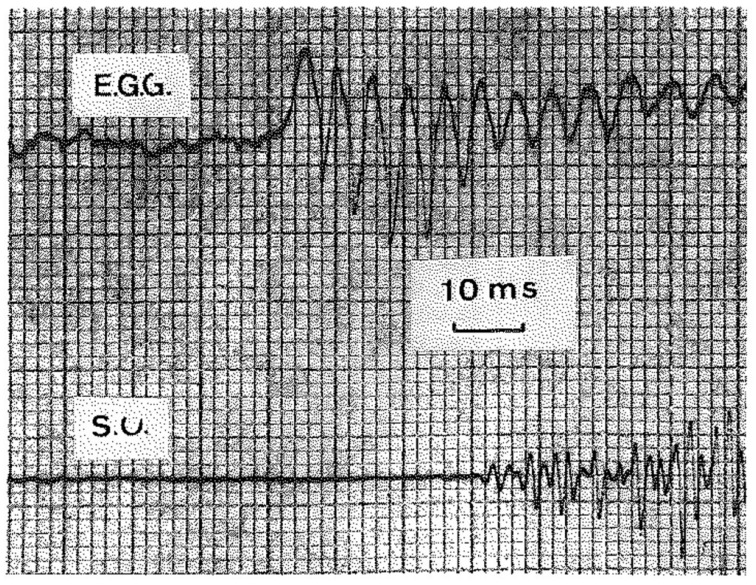
Attack of a sound of 175 Hz (horn). Simultaneous recording of EGG and sound oscillogram (SO). On the EGG signal, the maximum of VF contact is below, the minimum is above. Five glottal cycles precede the emission of sound. The amplitude of glottal movements is larger during these five first cycles than during the sustained emission. The glottal frequency of the first cycles is slightly higher than that of the sound to be played, but the adjustment is very quick.

**Figure 24 bioengineering-12-00155-f024:**
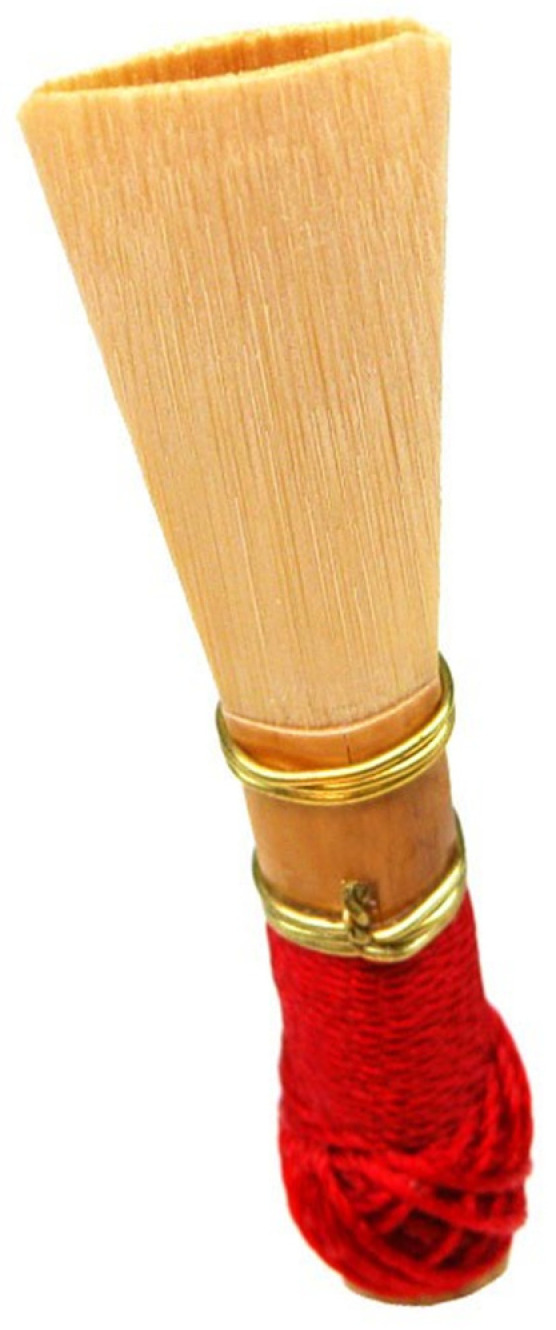
Vibratory double-reed of a tenor crumhorn, with its typical spindle shape. When the instrument is played, a cap covers this double reed, preventing any contact with the player’s lips.

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
