# Peer review of "Biophysics of Voice Onset: A Comprehensive Overview"

_bioengineering, 2025, doi:10.3390/bioengineering12020155_

Round 1
Reviewer 1 Report
Comments and Suggestions for Authors
I am uncomfortable with this paper and am critical of it in several ways. However, this is a review paper and is based in large part on work already published successfully by these author. I am not fully up to date on advances that have been made in the field in the last 30 years or so. Therefore, I am not confident about my criticisms but I will express them anyway.
I am a bit uncomfortble that the distinctions between actual physical parameters and estimates of them obtained by indirect means are blurred. This shows up first in the first figure presenting waveforms, Figure 10, and it persists in all the subsequent figures showing such waveforms.. Two of the plots are labelled Flow and Glottal Area. These plots are correctly identified in the caption as FGG (flow glottogram) and PGG (photoglottogram or transillumination). FGG is based on flow and sound waveforms recorded at the mouth. It is calculated by using the spectrum of the recorded sound waveform to invert the effects of vocal-tract acoustics on the flow waveform. There remains some doubt as to how accurately this can be done. PGG is an indirect measure obtained by illuminating the trachea from the neck and detecting how much of the light passes through the glottis. There is also some doubt as to how accurately the PGG represents glottal area, especially since the glottis doesn't have vertical walls, so it is unclear how "glottal area" should be measured.
Calculation of intraglottal pressure is is not well explained. Section 2.7 The description in Section 2.7 states that it is calculated from applying the Bernoulli equation to the particle velocity of airflow through the glottis. But particle velocity is calculated by dividing volume velocity by glottal area, which varies across the depth of the glottis. Therefore, intraglottal pressure varies over the depth of the glottis. In Section 5.1 (in the Biomechanics section), the explanation about use of the Bernoulli equation is changed, stating that it no longer applies when the glottis has a divergent shape along the vertical dimension. In that case, it is said that turbulence occurs and the relevant equation is the inertance equation. This leads to the conclusion, in section 5.3, that turbulence "can be assumed to be the triggering factor" in a soft(open) onset of phonation. This is the major conclusion of the paper.
I am concerned that a whole body of work on the significance of the vertical phase difference of glottal vibration has been ignored. The mathematical two-mass model of Ishizaka and colleagues, in which only the Bernoulli equation is used to calculated pressure produces oscillation, and excised larynges can be made to phonate without attachment to a vocal tract when placed in an appropriate prephonatory configuration and airflow with an appropriate subglottal pressure is supplied. I suggest that this should be addressed in this paper.
Finally, the vertical phase difference could be due to wave motion in the mucosal layer of the vocal folds, similar to waves on the surface of water when the wind blows across it. I don't believe that this requires turbulence.
Comments on the Quality of English LanguageBelow are suggestions for corrections of English language and other errors:
lines 143-144: "skewing to the right of the airflow curve" -> "skewing of the airflow curve to the right"
lines 157 and 158: two "e.g."s. Delete the 2nd one?
line 158: "fricative" -> "voiceless fricative"
line 166: "by a just" -> "by just"
line 168: "convicing" -> "convincing"?
line 208: "in own" -> "in our"
line 230 and 240: -> the term "soft I" has not been introduced. Should this be "soft (c)"?
line 250: "8ehavior" -> "behavior"
line 284: delete the last character (mu)
lines 325-326: "glottal duct is convergent glottis, that is to say upstream of the glottal narrowing" -> "glottal duct is convergent along the vertical dimension, meaning that it is narrower on the upstream side."
lines 326-327: "a divergent glottal duct, that is to say downstream of the narrowing," -> "a divergent glottal duct, meaning the duct is narrower on the downstream side,"
lines 336: "emission" -> "production"
line 353: "if" -> "is"
line 358: "being than" -> "being greater than" or "being more than"
line 399: "an transfer" -> "a transfer"
line 367: "successive" -> "successive cycles"
line 368: fix the size of the open bracket.
line 391: delete " glottal impedance,". (It isn't shown.)
lines 389 and 39: "30" - this refers to reference 30 and should be superscripted
line 393: "Calibration is made on images." what does this mean?
line 405: "mass VFs" -> "mass of the VFs"
line 422: "on the two sides of the oscillation axis" -> "around their oscillation axis"
lines 430-431: "atmospheric pressure at the glottal level is approximately normal" -> "pressure at the glottal is approximately at atmospheric level"
line 450: "more differentiated" is a poor description. Maybe "peakier"?
line 452: is "downward movement" meant to describe the EGG signal or the vocal folds. If the former, move it to just after "the electrical impedance undergoes a change first"
line 488: fix the font size of "3"
line 493: "Slight progressive decrease" -> "There is a slight progressive decrease"
line 499: delete "a" in "instances of a hard onset"
line 529: "an just" -> "and just"
line 727: fix this reference
Reviewer 2 Report
Comments and Suggestions for Authors
Comments and suggestions:
· The abstract lacks a clear statement of the paper's objectives and methodology. It jumps directly into findings without setting the context. I strongly recommend modifying the abstract.
· The introduction (Section 1) needs a clearer structure. It mixes background information with methodology details, which can be confusing for readers.
· The paper lacks a dedicated methodology section. While various methods are described throughout, a consolidated methodology with step-by-step explanations of each module is needed with clarity.
· There's no clear discussion section that synthesizes the findings and relates them back to existing literature or broader implications.
· Some figures (e.g., Fig. 2, 3, 4) lack scale bars or measurements, which could be helpful for quantitative comparisons, and the paper needs more quantitative data to support qualitative observations, especially in describing different onset types.
· Line 35: "To some extent, the soft voice onset reflects the voice offset." - This statement needs clarification or supporting evidence.
· Line 38: "The traditional videostroboscopic observation, as used clinically, is obviously not suited for analysing the characteristics of the onset phase." - The word "obviously" should be avoided in scientific writing unless the statement is truly self-evident.
· Line 97-98: "Time delay correction and calibration procedures have been described previously" - A specific reference should be provided here.
· Equation 1 (Line 139) is not properly formatted and lacks a clear explanation of all variables.
· Line 150: "As demonstrated by PGG, EGG, FGG, video laryngoscopy, high-speed film and VKG" - These abbreviations should be spelled out at first use.
· Some figure captions (e.g., Fig. 3, 4) lack sufficient detail to stand alone without referring to the main text.
· Extension of the literature will be appreciated to discuss in the literature section “An efficient violence detection approach for smart Cities surveillance system” and “VD-Net: An Edge Vision-Based Surveillance System for Violence Detection.”
Comments on the Quality of English Languagemoderate
Reviewer 3 Report
Comments and Suggestions for Authors
REVIEW: "Biophysics of Voice Onset: a Comprehensive Overview"(Bioengineering, 2024)
The article "Biophysics of Voice Onset: a Comprehensive Overview" provides an in-depth analysis of the mechanisms and dynamics involved in voice onset, exploring the physiological, aerodynamic, and biomechanical aspects of vocal fold oscillation. It presents a detailed examination of different types of vocal onsets (soft, breathy, and hard) using advanced imaging and measurement techniques, such as high-speed video, electroglottography, and photoglottography. A significant strength of the article is its comprehensive approach to combining multiple methodologies, offering a nuanced understanding of vocal onset dynamics.
However, the paper's dense technical language and extensive detail may hinder readability for a broader audience in Bioengineering. Additionally, the abstract and conclusion could be more focused to clearly highlight the study's contributions and implications, especially for Linguistics and the study of human language. Overall, the article makes a valuable contribution to the field of voice biophysics by synthesizing complex concepts and experimental findings.
I recommend this paper for publication in Bioengineering only after major revisions to improve clarity, depth of analysis, and, especially, ethical concerns:
The study does not provide sufficient details on whether participants represent all genders and age groups, which could introduce a gender or age bias if the sample is skewed. The article primarily references "normal male trained vocalists" and "young female singing students," suggesting a potential imbalance in gender representation that might affect the generalizability of the findings. “Normal” is a man? Additionally, it mentions using these data from participants in other studies, but there is no indication that the research underwent an ethics committee review, nor is there explicit mention of informed consent for biometric data collection. To address potential biases and legal concerns, the authors should clarify the demographic diversity of their sample and ensure that ethical procedures, including informed consent and committee approval, are properly documented. This is a key point of the review.
The article also should:
- Clarify Linguistic Relevance: Explain how the biophysical findings relate to phonetic and phonological concepts, such as voice onset time (VOT) and articulation. This connection will make the study more accessible and relevant to linguists.
- Expand the Discussion Section: Relate findings to linguistic phenomena, such as speech disorders, prosody, or language acquisition, and compare how the physical onset dynamics impact these areas.
- Include Examples and Applications: Providing concrete examples of how voice onset influences speech sounds across different languages and in language disorders will make the review more complete and relatable to linguists interested in cross-linguistic phonetic variations.
These improvements will help make the article more engaging and relevant for the linguistic community, encouraging interdisciplinary dialogue.
Minor Comments:
-
Author Name Issues: The name of the second author "jean.lebacq" is not properly capitalized and formatted in the title section (line 3). Should it be "Jean Lebacq."?
-
Formatting Errors:
-
The title "Review" should be centered or formatted consistently with journal guidelines.
-
Several instances of spacing issues are evident, for example, between "electro-" and the next line in the abstract.
-
The citation formatting needs improvement, especially in "References" section. Consistency in citation style is essential.
-
-
Typos and Grammatical Issues:
-
Page 2, Line 16: A space is needed before the colon in "plateau :". Please review these details in the main article.
-
Replace instances of "onset may be 'breathy'" with consistent formatting, either using single or double quotes throughout the text.
-
-
Figure Captions: The figure captions should be uniform and self-explicative. Some captions are complete sentences, while others are not or referred to previous figures. Ensure all captions are either in sentence format with a period or a title format without one, and clarify if are from a men or a woman, and the age.
Major Revisions:
Abstract: The abstract requires significant improvement to accurately reflect the article's content. It currently lacks information on the review done, methodologies and key findings. Include a brief explanation of the primary objectives, methods, and major conclusions. Sometimes I can find this information in introduction (lines 39-41, 57, for example).
-
Introduction:
-
The introduction gives a good background on vocal onset, but it should more clearly define the article's objectives and impact in Linguistics.
-
It is not clear how: “Combining techniques of imaging with physical methods, it becomes possible to obtain a detailed insight, qualitative and quan titative, into the essential aspects of the vocal onset” (lines 44-46) leads to the subsequent list of items.
-
Clarify the significance of the study and its contribution to the field, particularly how it extends previous research. In this regard, some classical references on onset are missing, especially on a key missing topic as Glottal Pulse Shape (i.e. see seminal work by Rosenberg, 1971 Effect of Glottal Pulse Shape on the Quality of Natural Vowels (uva.nl) and late references)
-
Please clarify if the study includes women, men and children, with their consent and ethical commitee aproval.
-
-
Relevant investigation methods
-
The article details multiple investigation methods, but the descriptions could be more concise. Consider restructuring this section to focus on how each method contributes to understanding the voice onset phenomenon.
-
Include a flowchart or table summarizing the different techniques used, as this would enhance readability.
-
-
Systematization Of The Voice Onset Categories
-
Several sections are very dense with technical information, such as this section. Break down complex concepts into simpler language or provide additional explanations for clarity: a table summarazing this section may help.
-
-
Morphological Analysis
-
Very interesting section, but again I think a table comparing hard, soft and breathy onset, is mandatory. Otherwise, the discussion section should be expanded to compare these findings with existing literature more thoroughly.
-
Biomechanics
-
In the "Biomechanics" section, the article presents several formulas, such as the Bernoulli equation and the inertance equation, which are crucial for understanding the dynamics of vocal fold oscillation. However, these formulas are not consistently connected with the data presented in the figures, particularly Figure 14, which shows the phase lead progression of intraglottal pressure relative to the glottal area, or Figure 16, where equations between varables are missing. For instance, the article could improve clarity and scientific rigor by providing the best-fitting function or equation that models the data in Figure 14, demonstrating how the theoretical formulas align with the observed experimental results. This integration would help reinforce the relationship between the biomechanical theory and the empirical findings, making the analysis more comprehensive and accessible.
-
Another question: How can indirectly ‘the energy of a sound’ be obtained with this measures? This is a good question that can open the study to linguists.
-
Improve axis in figure 19 indicating clearly magnitudes and units, also in subfigures.
-
Here and in other sections, authors overlook the extensive existing literature on the phenomenon of coarticulation, which is highly relevant to voice onset dynamics. Notably, they do not reference the foundational work of Paul Menzerath, who explored the principles of coarticulation and its impact on speech production in his studies. Menzerath’s works provides crucial insights into how adjacent speech sounds influence one another during articulation, a factor that could significantly affect the biomechanical aspects of voice onset analyzed in this study. Including Menzerath's research would offer a more comprehensive understanding of how coarticulation contributes to the dynamics of voice onset in natural speech contexts.
-
References:
-
Menzerath, P., & de Oleza, J. M. (1928). Spanische Lautdauer: Eine experimentelle Untersuchung. Bonn: Peter Hanstein Verlag.
-
Menzerath, P., & Lacerda, A. (1933). Koartikulation, Seuerung und Lautabgrenzung. Berlin: Walter de Gruyter & Co.
-
Frequency Control In Onset Of Singing Intervals
-
Line 505: teachers are men or woman, or both? Please clarify.
-
Authors claim (539-542):”This phenomenon may be attributed to a necessary mechanical readjustment and stabilisation of muscle tension, as well as a new equilibrium between glottal resistance and subglottal pressure.” Any reference to support this claim?
-
Conclusion
-
The conclusion adequately summarizes the findings, but it would benefit from emphasizing the study's broader implications and potential applications in linguistics: how can this work improve language disorders diagnostics?
-
Can authors relate their work with other well-known dynamic phenomena such as coarticulation?
-
-
Highlight the article's contributions and suggest future research directions. This will provide a better sense of the study's implications.
-
References
-
Ensure all citations are complete and follow a consistent format. Some references lack volume or page numbers. Review how to cite webpages.
-
The article exhibits an excess of self-citations, which could limit the perspective and perceived objectivity in the review of the voice onset phenomenon. Ate all necessary? Please review it. Additionally, it overlooks classic works on the topic, such as those by Paul Menzerath, among others, that have significantly contributed to the understanding of coarticulation and voice onset processes. Including a more comprehensive and balanced review of the existing literature, incorporating both historical and contemporary studies, would enrich the theoretical foundation of the article and strengthen its connection to the established knowledge in the field.
-
By addressing these points, the article can achieve a more polished and coherent presentation, making it suitable for publication in Bioengineering.
Round 2
Reviewer 1 Report
Comments and Suggestions for Authors
My previous comments have generally been dealt with adequately.
Additional minor comments:
Line 44: "is the pendant to". Not clear what his means. Please find another way of stating this.
Line 49: add commas after "uttering" and "conditions"
Line 71: "sound production." -> "sound."
Line 78: delete "photoglottagr videolaryngostroboscopy"
Line 79: "Droppler" -> "Doppler"
Line 87: "laryngeal vocal tract" -> "vocal tract"?
Line 97: I don't understand the meaning of "line frame/s" or of the second set of numbers.
Line 120: "[6, 7]in" -> "[6, 7] in"
Line 157: "v2". The 2 needs to be superscripted.
Line 158: "In" -> "in"
Line 159: "proposed ]graphically" -> "proposed"?
Line 161: "VF movement" -> "VF area"
Line 173: "film" -> "filming"
Line 214: superscripted "27" -> "[27]". Similar commets apply to lines 329, 383, 386, 509, 581.
Line 344: The superscripted "3" should probably be deleted
Line 527: The "cycle 3" with supercripted 3 should probably be "cycle 3" with 3 as normal text.
Lines 357,358: I think "upstream" and "downstream" should be reversed. "upstream " means closer to the source and "downstream" means "in the direction of flow".
Line 427: "example of record" -> "example of a record"
Line 468, 469: "atmospheric". Is this the proper word in either or both of these places?
Figures 17 and 19: These needs x-axis labels, "cycle number".
Line 531: "17" -> "18"
Figure 20, top: "10x" -> "9x". Does "nl" mean "normal"?
Figure 21: Are the length of the line segments for "5 10 cycles" correct?
Line 755: Is the parts after the colon correct"
Comments on the Quality of English LanguageSee above.
Author Response
Thanks for pointing out typographical errors! Corrections have taken place.
According to a comment of Reviewer 1, Fig. 20 has been changed (recoloured): It shows 10 repetitions of a voice onset, but the reviewer could indentify only 9 tracings. This is right and is due to the fact that in the original picture, each tracing has a different colour, which is actually not necessary to understand the meaning of the figure. One of the tracings had a very pale colour and became almost invisible. Now all 10 tracings are simply black.
Thanks to the reviewer for pointing out an error with ‘upstream’ and ‘downstream’. Terms were reversed.
Reviewer 2 Report
Comments and Suggestions for Authors
The responses and actions taken in the paper are not satisfactory; just highlighting is not enough, and they should clearly mention and write responses to each point and action, such as "what they did in the paper," according to the reviewer's concern.
Comments on the Quality of English Languageneeded
Author Response
Clear statement about the paper’s objective : this is now clearly stated in the Introduction as well as in the Abstract : “The objective of this article is…” and “ The aim of this article is to…
Methodology : the methodological issues were grouped in Section 2 : the direct measures as well as those resulting from a computed combination of two direct measures. All necessary references are provided.
Some formulations that could be perceived as unclear (e.g. ‘To some extent the soft voice onset reflects the voice offset’) have been rewritten (e.g. ‘To some extent the soft voice onset is the inverted homologue of the voice offset’).
Discussion and findings : the three key-findings are now clearly individualised, in the Abstract as well as in the Conclusion. These key-findings are discussed in the different sections, and - there where relevant - a specific discussion with the relations to the existing literature is referred to. These references mainly concern our own original articles used for constructing the present comprehensive survey.
Some small changes have been made in the captions of the figures.
Typographical errors have been corrected.
Reviewer 3 Report
Comments and Suggestions for Authors
Authors should recognize (gender, age, ethnic...) biases in their study and Ethical aproval is needed, as well as information consent signed by all participants. It is mandatory in biometry in EU reasearch with humans.
Author Response
Even if we agree that our biophysical findings could probably be connected to phonetic, phonological and linguistic phenomena (prosody, language acquisition etc.), and be of some relevance to linguists, we do not feel qualified to concretely discuss such links and to illustrate them by adequate examples.
The limitations of our findings (to modal emissions and with limited numbers of subjects) are clearly stated in the conclusions.
However, we explicitly mention that we hope that researchers in linguistic fields can use our findings for making links with their own research topics.
The comments on ethical approvement and informed consent have been addressed
Round 3
Reviewer 3 Report
Comments and Suggestions for Authors
Although I regret that the authors have not addressed some of my comments, specifically those regarding the inclusion of Menzerath's seminal works and the connection of their work with linguistics, which would have broadened the scope of their review, I understand that they have made the minimum changes to be published in this journal and field. I encourage them to do so in the future, as well as to consider requesting written informed consent from participants in their biometric studies, before carrying them out. Congratulations on your work.
Author Response

(The authors gave the same response as above.)
